# ⚒ FORGE4D: FEED-FORWARD 4D HUMAN RECONSTRUCTION AND INTERPOLATION FROM UNCALIBRATED SPARSE-VIEW VIDEOS

## ABSTRACT

Instant reconstruction of dynamic 3D humans from uncalibrated sparse-view videos is critical for numerous downstream applications. Existing methods, however, are either limited by the slow reconstruction speeds or incapable of generating novel-time representations. To address these challenges, we propose *Forge4D*, a feed-forward 4D human reconstruction and interpolation model that efficiently reconstructs temporally aligned representations from uncalibrated sparse-view videos, enabling both novel view and novel time synthesis. Our model simplifies the 4D reconstruction and interpolation problem as a joint task of streaming 3D Gaussian reconstruction and dense motion prediction. For the task of streaming 3D Gaussian reconstruction, we first reconstruct static 3D Gaussians from uncalibrated sparse-view images and then introduce learnable state tokens to enforce temporal consistency in a memory-friendly manner by interactively updating shared information across different timestamps. For novel time synthesis, we design a novel motion prediction module to predict dense motions for each 3D Gaussian between two adjacent frames, coupled with an occlusion-aware Gaussian fusion process to interpolate 3D Gaussians at arbitrary timestamps. To overcome the lack of the ground truth for dense motion supervision, we formulate dense motion prediction as a dense point matching task and introduce a self-supervised *retargeting loss* to optimize this module. An additional occlusion-aware *optical flow loss* is introduced to ensure motion consistency with plausible human movement, providing stronger regularization. Extensive experiments demonstrate the effectiveness of our model on both in-domain and out-of-domain datasets.

## 1 INTRODUCTION

Instant 4D human reconstruction from uncalibrated sparse-view video streams is essential for various application scenarios, including real-time livestreaming (Xu et al., 2020), sports broadcasting, augmented/virtual reality (AR/VR) (Carmigniani & Furht, 2011), articulation modeling (Chen et al., 2023; Guo et al., 2025; Liu et al., 2025a; Zhang et al., 2025a), and immersive holographic communication (Tu et al., 2024). However, this task remains challenging due to the inherent difficulty of simultaneously recovering accurate human body geometry and dense motion trajectories from unposed sparse-view video streams, while maintaining the real-time interactivity required for practical applications. For example, holographic communication systems demand high interactability, while sports broadcasting requires the ability to present novel views at any time for enhanced viewing experiences and precise evaluation of athletic performance.

Existing works (Zhang et al., 2024; Li et al., 2024b; Jiang et al., 2024) typically rely on iterative optimization over entire dense-view video sequences for each scene. These approaches depend heavily on calibrated camera parameters and suffer from prolonged training durations required for 4D representation convergence. Meanwhile, recent advances in large-scale visual geometry models (Wang et al., 2025a; 2024a; 2025d) have enabled intermediate 3D point cloud reconstruction and camera pose estimation from arbitrary long uncalibrated image sequences in a feed-forward manner. However, the inherent limitations of point cloud representations restrict their ability to achieve photorealistic novel view synthesis. Subsequent works (Jiang et al., 2025; Ye et al., 2024) have extended feed-forward reconstruction models to predict static 3D Gaussians, enabling photorealistic

novel view synthesis. Nevertheless, these methods remain incapable of handling dynamic scenes and synthesizing novel-time images.

In this work, we propose Forge4D, the first *feed-forward* model for 4D human reconstruction that enables *novel-view* and *novel-time* synthesis from fixed multi-view *uncalibrated sparse-view* videos in an *efficient streaming manner*. Our framework enables: 1) efficient reconstruction of temporally consistent 3D Gaussian assets from streaming sparse-view video inputs, and 2) accurate frame-wise dense 3D motion prediction for human subjects and 4D interpolation for novel-time synthesis. To achieve these goals, we decompose the 4D reconstruction and interpolation problem into two tasks: streaming 3D Gaussian reconstruction and dense human motion prediction. This design offers two advantages: 1) it simplifies the problem for feed-forward regression, and 2) the reconstructed streaming 3D Gaussians provide visual supervision for accurate dense motion prediction. Specifically, for streaming 3D Gaussian reconstruction, we leverage the pretrained knowledge prior from the large 3D reconstruction model VGGT (Wang et al., 2025a) and adapt it to predict streaming key-frame 3D human Gaussian assets. This adaptation is non-trivial due to two major challenges. First, a scale discrepancy exists between VGGT's output and the real-world metric scale inherent in ground-truth camera extrinsics, causing fundamental misalignment and unstable optimization with novel view photometric loss. Second, naively feeding VGGT with multiple video frames suffers from long reconstruction duration, low interactivity, and out-of-memory (OOM) issues due to increasing image tokens for global attention. To address the scale issue, we propose to maintain a *metric gauge* and force the model to generate a temporally consistent scale in Sec. 3.2, which not only improves the stability under novel view supervision, but also enables metric measurement. For the efficiency and OOM problem, we decompose the spatial and temporal dimensions of sparse-view videos and propose state tokens in Sec. 3.3 to iteratively incorporate temporal information in a streaming manner.

For dense human motion prediction and novel-time synthesis, we propose a dense human motion prediction module in Sec. 3.4 to facilitate 3D representation synthesis at arbitrary intermediate timestamps. In contrast to prior approaches that depend on merely middle-frame photometric supervision, the key insight of our approach is that we formulate the task of dense motion prediction as a 3D Gaussian point-matching problem. However, there is no ground truth dense 3D human motion for supervision of this module. Therefore, we propose a novel *retargeting loss* that projects current 3D Gaussians to adjacent frames with the predicted dense motion and supervises the rendered images against ground truth. This regularization optimizes the dense motion in a self-supervised manner. For a stronger regularization, we also propose an *occlusion-aware optical flow loss*, which projects the 3D dense motion into 2D optical flows and explicitly aligns them with optical flows from a prior model to enhance the plausibility of predicted human motions. Given the dense motion between two timestamps, we deform the dynamic 3D Gaussians from the two nearest frames under a constant velocity assumption. These deformed representations are then merged using a lightweight fusion MLP that explicitly accounts for occlusion through a dual matching mechanism. Experimental results on benchmark datasets demonstrate the efficiency and effectiveness of the proposed framework.

The main contributions of this work are summarized as follows:

- We propose the first feed-forward model for 4D human reconstruction in real-world metric scale from uncalibrated sparse-view videos, enabling novel view synthesis and novel-time 4D interpolation in an efficient streaming manner.

- Our model simplifies this task by decomposing it into subsequent streaming 3D Gaussian prediction and dense human motion estimation tasks. The novel *metric gauge* regularization, *retargeting loss*, and *occlusion-aware optical flow loss* stabilize the optimization and significantly improve motion prediction, photorealistic novel-view and novel-time synthesis.

- We introduce a novel motion-guided, occlusion-aware Gaussian fusion method for 3D Gaussian interpolation, enabling novel-time synthesis and effectively mitigating flickering and jittering artifacts caused by temporal redundancy in dynamic 3D Gaussian representations.

## 2 RELATED WORK

**Dynamic Scene Reconstruction and Streaming.** Dynamic scene reconstruction from multi-view videos is crucial for numerous real-world applications. Prior methods primarily focus on optimizing a unified 4D representation to match dense multi-view 2D observations either by incorporating tem-

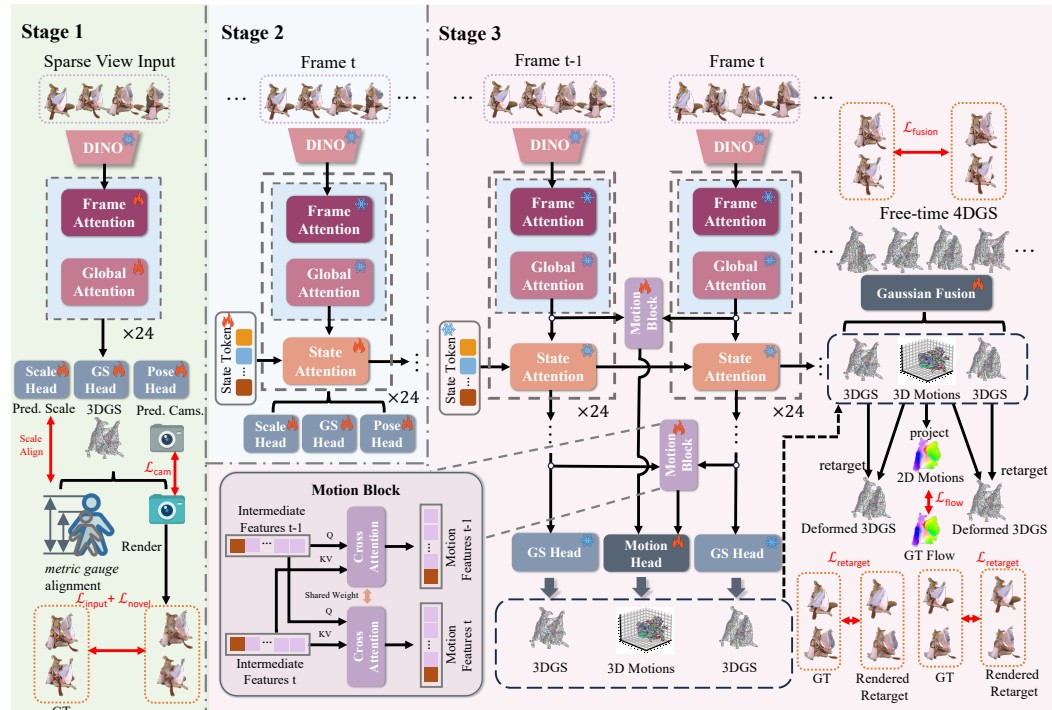

Figure 1: The overall pipeline of *Forge4D*. It is trained in three stages: (1) static feed-forward 3D Gaussian reconstruction stage; (2) a streaming stage temporally aligned via state tokens; and (3) a feed-forward 4D reconstruction stage that predicts dense motion for each 3D Gaussian and interpolates free-time 3D Gaussians using an occlusion-aware fusion process.

poral dimensions into spatial coordinates (Jin et al., 2025; Zhang et al., 2024; Wu et al., 2024; Duan et al., 2024; Lee et al., 2024) or by deforming 3D representations from keyframes using dynamic factors (Luiten et al., 2024; Lin et al., 2024; Jiang et al., 2024; Wang et al., 2025c; Sun et al., 2024b). Another line of research assumes causal inputs and reconstructs per-frame 3D representations in a streaming manner (Girish et al., 2024; Liu et al., 2025c; Yan et al., 2025; Sun et al., 2024a). However, these methods suffer from a limited reconstruction speed and are sensitive to the number of input views. In contrast to these iterative optimization-based approaches, *Forge4D* introduces an efficient feed-forward model that reconstructs the entire 4D scene in a single forward pass from uncalibrated sparse videos, significantly enhancing interactivity and applicability to downstream tasks.

**Feed Forward Reconstruction.** Recent advances in visual geometry models have demonstrated the capability of deep neural networks for 3D reconstruction from multi-view images in a feed-forward manner. DUSt3R (Wang et al., 2024a), VGGT (Wang et al., 2025a), and $\pi^3$ (Wang et al., 2025d) enable direct regression of camera poses and 3D point maps in the first frame's coordinate space. To enable photorealistic novel view synthesis, another line of works (Charatan et al., 2024; Chen et al., 2024b; Zheng et al., 2024; Hu et al., 2024; Chen et al., 2024a; Tu et al., 2025) directly predict static 3D Gaussians (Wang et al., 2024c; Shen et al., 2024; Yi et al., 2024; Xu et al., 2025a; Zhang et al., 2025b; Liu et al., 2025b) or textured meshes (Li et al., 2024a) from calibrated multi-view images. To alleviate the reliance on camera calibration, NoPosplat (Ye et al., 2024) and AnySplat (Jiang et al., 2025) propose to reconstruct 3D Gaussians from uncalibrated multi-view images. However, all these methods are limited to per-timestamp static reconstruction and cannot synthesize novel-time 3D representations. Although L4GM (Ren et al., 2024) extends the static Gaussian reconstruction framework (Tang et al., 2024) to feed-forward 4D reconstruction, it suffers from low-resolution reconstructions and poor generalization on real-world subjects. Concurrent to our work, recent methods (Xu et al., 2025b; Lin et al., 2025b;a) target 4D Gaussian reconstruction from monocular calibrated videos, and StreamSplat (Wu et al., 2025) further extends to uncalibrated ones. However, these methods neither explore the critical connection between 4D reconstruction and 3D point matching nor formulate motion learning as an explicit geometric correspondence problem, resulting in suboptimal performance. In contrast, our model is specifically designed to recover detailed geometry, appearance, and dense frame-wise motion for human performance from uncal-

162
163
164
165
166
167
168
169
170
171
172
173
174
175
176
177
178
179
180
181
182
183
184
185
186
187
188
189
190
191
192
193
194
195
196
197
198
199
200
201
202
203
204
205
206
207
208
209
210
211
212
213
214
215

ibrated multi-view videos. By reformulating 4D reconstruction as a 3D Gaussian point matching task and introducing specialized losses for motion retargeting and occlusion-aware flow alignment, our approach achieves superior reconstruction quality and temporal consistency.

# 3 METHOD

## 3.1 OVERVIEW

This work utilizes a transformer-based model $\mathcal{D}_{4D}$ for feed-forward reconstruction and novel-time interpolation of dynamic 4D Gaussian $\mathcal{G}_{4D}$ from $n$ sparse uncalibrated videos $\{\boldsymbol{I}_i^t\}_{i=0,t=0}^{n,k}$ with a consistent video length of $k$ and no camera motion, which can be expressed in the form of:

$$\mathcal{G}_{4D} = \mathcal{D}_{4D}(\{\boldsymbol{I}_i^t\}_{i=0,t=0}^{n,k}). \tag{1}$$

However, the strong entanglement between object geometry and motion makes the direct regression of 4D Gaussians challenging. To address this issue, we propose to decompose the problem into a streaming 3D Gaussian reconstruction task and a dense motion prediction task, and develop a progressive training pipeline. As shown in Fig. 1, the proposed pipeline is composed of three stages: 1) a feed-forward static 3D reconstruction stage to reconstruct static 3D Gaussians in the real-world metric scale, 2) a streaming dynamic reconstruction stage to reconstruct streaming 4D Gaussians in an efficient and memory-friendly way, and 3) a dense motion prediction and Gaussian fusion stage to enable novel-time synthesis. In the first stage, a novel metric gauge calculation method is proposed to align the backbone output scale with the real-world scale, which is critical to more stable supervision of novel views. While directly applying the 3D Gaussian reconstruction pipeline to each time stamp suffers from scale misalignment between different times, the main purpose of the second stage is to align different scales across different time stamps. To this end, we propose to use a state token to encode information from former frames and interact with the immediate frame efficiently, while being memory-friendly. In the final stage, a novel motion prediction module is proposed, together with a novel dual frame *retargeting loss* and *occlusion-aware optical flow loss* that marry the task of dynamic Gaussian motion prediction to the task of point matching. To enable novel-time synthesis, an additional occlusion-aware Gaussian fusion procedure is proposed for better dynamic 3D Gaussian interpolation and to resolve the jittering and flashing problem.

## 3.2 STAGE 1: FEED-FORWARD 3D GAUSSIAN RECONSTRUCTION

Recent advances in large 3D reconstruction models have demonstrated remarkable capabilities in recovering colored point maps and camera poses from a set of uncalibrated images. However, the point cloud representation limits the capability for photorealistic synthesis. To address this issue, we propose a feed-forward 3D Gaussian reconstruction model $\mathcal{D}_{3D}$ that leverages the geometry prior within these foundations, while introducing a 3D Gaussian prediction branch for photorealistic rendering. Specifically, we use a pre-trained VGGT as our backbone and predict pixel-aligned 3D Gaussians with an additional DPT (Ranftl et al., 2020) head as $\mathcal{G}_{3D}^t = \mathcal{D}_{3D}(\{\boldsymbol{I}_i^t\}_{i=0}^n)$, where $\mathcal{G}_{3D}^t = \{\boldsymbol{P}_i^t, \boldsymbol{O}_i^t, \boldsymbol{C}_i^t, \boldsymbol{Q}_i^t, \boldsymbol{S}_i^t\}_{i=0}^n$, with $\boldsymbol{P}_i^t \in \mathbb{R}^{3\times H\times W}$, $\boldsymbol{O}_i^t \in \mathbb{R}^{1\times H\times W}$, $\boldsymbol{C}^t \in \mathbb{R}^{3\times H\times W}$, $\boldsymbol{Q}_i^t \in \mathbb{R}^{4\times H\times W}$, and $\boldsymbol{S}_i^t \in \mathbb{R}^{3\times H\times W}$ representing the Gaussian position, opacity, color, rotation, and scale attribute maps, respectively, from view $i$ of size $H \times W$ at time $t$.

To supervise this branch, photometric losses (e.g., L2, SSIM, LPIPS) are applied between the rendered and ground-truth (GT) images. However, a fundamental scale ambiguity occurs between the output scale of VGGT (i.e., normalized point clouds) and the real-world metric scale. Directly applying photometric supervision without addressing this scale discrepancy results in an unstable optimization trajectory and fails to converge to a coherent 3D structure, as further evaluated in Sec. 4.3. To resolve this issue, we introduce a *metric gauge* regularization term $p_{\text{gauge}}$ to align the scale of the GT novel-view camera extrinsics with the model's internal coordinate system, thereby stabilizing training. This approach is grounded in the key insight that if the model's predicted camera poses

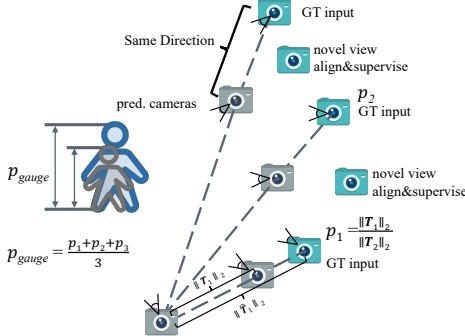

Figure 2: Gauge illustration.

and intrinsics are accurate, their difference from the GT poses should be primarily a consistent translation scaling factor, as visualized in Fig. 2. Formally, for $n$ input cameras, we calculate the ratio $p_i = \|\boldsymbol{T}_i\|_2 / \|\hat{\boldsymbol{T}}_i\|_2$ of translation magnitudes between each predicted camera and its GT counterpart. The novel view cameras are then scaled using the mean of these ratios, $p_{\text{gauge}} = \frac{1}{n-1}\sum_{i=1}^{n} p_i$. This factor is also utilized for scale head supervision, which predicts $\hat{p}_{\text{gauge}}$ for metric scale recovery during evaluation. To ensure that the metric gauge accurately represents the scale difference and to simultaneously refine the predicted camera parameters, we propose a comprehensive camera loss:

$$\mathcal{L}_{\text{cam}} = \sum_{i=0}^{n} \|\boldsymbol{q}_i - \hat{\boldsymbol{q}}_i\|_2 + \sum_{i=1}^{n} \left\| \frac{\boldsymbol{T}_i}{\|\boldsymbol{T}_i\|_2} - \frac{\hat{\boldsymbol{T}}_i}{\|\hat{\boldsymbol{T}}_i\|_2} \right\|_2 + \sum_{i=1}^{n} |p_i - p_{\text{gauge}}| + |\hat{p}_{\text{gauge}} - p_{\text{gauge}}|, \quad (2)$$

which supervises the model for accurate rotation $\boldsymbol{q}_i$, translation direction, consistent relative scaling, and the scale prediction header. Our full training objective then combines this metric-aware camera loss with multi-view photometric supervision. The photometric loss $\mathcal{L}_{\text{input}} = \sum_{i=0}^{n}(\|\boldsymbol{I}_i - \hat{\boldsymbol{I}}_i\|_2 + \lambda_{\text{SSIM}}\text{SSIM}(\boldsymbol{I}_i, \hat{\boldsymbol{I}}_i) + \lambda_{\text{LPIPS}}\text{LPIPS}(\boldsymbol{I}_i, \hat{\boldsymbol{I}}_i))$ is applied to the $n$ input views, and a corresponding loss $\mathcal{L}_{\text{novel}}$ is applied to $m$ held-out novel views, ensuring high-fidelity reconstruction across all perspectives. Thus, the total loss for our feed-forward 3D Gaussian reconstruction model is $\mathcal{L}_{\text{3D}} = \mathcal{L}_{\text{cam}} + \mathcal{L}_{\text{input}} + \mathcal{L}_{\text{novel}}$. Supervised by this combined objective, our model not only infers coherent 3D Gaussians from uncalibrated RGB inputs, but also enables metric measuring, which we discuss thoroughly in Appendix C.

### 3.3 STAGE 2: DYNAMIC HUMAN STREAMING WITH STATE-TOKEN GUIDED ALIGNMENT

In stage one, we obtain a feed-forward network for static 3D Gaussian reconstruction from sparse-view images. However, to apply for video scenarios, the output scale of this network is not aligned across timestamps, leading to temporal inconsistency. To address this, one vanilla way is to stack all sparse video tokens for global attention in VGGT, which results in OOM and low reconstruction speed issues. Instead, we decompose the spatial and temporal dimensions of sparse-view videos and introduce a state token to enforce the temporal consistency in an efficient and memory-friendly way. Specifically, we employ a learnable *state token* that iteratively encodes information from all previous frames and broadcasts it to the current frame. These tokens inject temporal information by serving as the Key and Value in a cross-attention layer applied to the current frame's features. Conversely, the token itself is updated by attending to the current frame's features, where it serves as the Query. To further enhance the temporal stability, we extend the metric gauge regularization in Sec. 3.2 to a temporal form. The global scale factor is now computed over all $n$ cameras and $k$ timestamps as $p_{\text{gauge}} = \frac{1}{(n-1)(k-1)}\sum_{t=1}^{k}\sum_{i=1}^{n} p_i^t$, which is also utilized to supervise the general gauge $\hat{p}_{\text{gauge}}^t$ prediction in the scale prediction header. Consequently, we generalize the camera loss to supervise the temporal cross-attention layers across the entire sequence as:

$$\mathcal{L}_{\text{cam}} = \sum_{t=0}^{k}\sum_{i=0}^{n} \|\boldsymbol{q}_i^t - \hat{\boldsymbol{q}}_i^t\|_2 + \sum_{t=0}^{k}\sum_{i=1}^{n} \left\| \frac{\boldsymbol{T}_i^t}{\|\boldsymbol{T}_i^t\|_2} - \frac{\hat{\boldsymbol{T}}_i^t}{\|\hat{\boldsymbol{T}}_i^t\|_2} \right\|_2 + \sum_{t=0}^{k}\sum_{i=1}^{n} |p_i^t - p_{\text{gauge}}| + \sum_{t=0}^{k} |\hat{p}_{\text{gauge}}^t - p_{\text{gauge}}|. \tag{3}$$

This ensures consistent camera rotation, translation direction, and global scale across time.

### 3.4 STAGE 3: DENSE HUMAN MOTION PREDICTION AND DYNAMIC GAUSSIAN FUSION

A general 4D representation should enable the synthesis of 2D images from arbitrary camera viewpoints at *any moment* in time. This requires the capability to interpolate the representation to intermediate timestamps beyond the input frames. In this work, we adopt dynamic 3D Gaussians as our fundamental 4D representation and assume a linear motion model that propagates 3D Gaussians between consecutive frames, following previous 4D Gaussian reconstruction works (Wang et al., 2025c). Mathematically, we formulate our 4D Gaussian representation as:

$$\mathcal{G}_{4D} = \{\{\mathcal{G}_{3D}^t\}_{t=0}^{k}, \{\boldsymbol{M}_{i,\{1,2\}}^t\}_{i=0,t=0}^{n,k}, \boldsymbol{F}_\theta\}, \tag{4}$$

where $\mathcal{G}_{3D}^t$ denotes the 3D Gaussian attribute maps at time $t$, $\boldsymbol{M}_{i,\{1,2\}}^t \in \mathbb{R}^{2\times3\times H\times W}$ represents the associated 3D motion field for view $i$, and $\boldsymbol{F}_\theta$ is a learnable fusion function that adaptively combines Gaussian attributes at novel timestamps in account of occlusion relationships.

To predict the 3D motion map $\boldsymbol{M}_{i,\{1,2\}}^t$, we introduce a dense motion prediction block that operates on the static streaming reconstruction described in Sec. 3.2. This block predicts a dense, pixel-

aligned dual motion field for each 3D Gaussian. Specifically, when processing a new frame at time $t$, the block infers: (1) a backward 3D motion $\boldsymbol{M}_{i,1}^t \in \mathbb{R}^{3 \times H \times W}$ that warps the current 3D Gaussians (time $t$) to the previous timestamp $t-1$, and (2) a forward 3D motion $\boldsymbol{M}_{i,2}^{t-1} \in \mathbb{R}^{3 \times H \times W}$ that warps the 3D Gaussians from the previous frame (time $t-1$) to the current frame $t$. This process is repeated symmetrically when the subsequent frames arrive. The motion prediction block comprises the same number of attention blocks as the backbone model. Each motion attention block takes as input the corresponding intermediate features from frames $t$ and $t-1$ produced by the backbone. The output features from all motion attention blocks are aggregated and passed to a motion DPT head to produce the final dual motion prediction $\boldsymbol{M}_{i,\{1,2\}}^t$.

Given 3D Gaussian assets at two successive frames $t$ and $t-1$, along with their corresponding motions, the 3D Gaussians at the middle timestamp $t'$ can be acquired by warping these two frames with a consistent velocity assumption. To be specific, for each 3D Gaussian in frame $t$, it is deformed to the middle timestamp by adding a displacement proportional to its temporal distance to time $t$. The 3D Gaussians in frame $t-1$ are also deformed to this middle timestamp $t'$ in the same way. This deforming process can also be represented as:

$$\boldsymbol{P}_i^{t \to t'} = \boldsymbol{P}_i^t + |t' - t| \cdot \boldsymbol{M}_{i,1}^t, \boldsymbol{P}_i^{t-1 \to t'} = \boldsymbol{P}_i^{t-1} + |t' - (t-1)| \cdot \boldsymbol{M}_{i,2}^{t-1}. \tag{5}$$

However, there are no ground truth dense human motions for supervision. To effectively train this block and the deformation process, we introduce a novel *retargeting loss* that optimizes the predicted 3D motion using 2D photometric constraints in a self-supervised manner. Specifically, for 3D Gaussians at time $t$, we deform their positions to time $t-1$ with the prediction dense motion as $\boldsymbol{P}_i^{t \to t-1} = \boldsymbol{P}_i^t + \boldsymbol{M}_{i,1}^t$, while retaining other attributes. This forms an intermediate 3D Gaussian representation $\mathcal{G}_{3D}^{t \to t-1} = \{\boldsymbol{P}_i^{t \to t-1}, \boldsymbol{O}_i^t, \boldsymbol{C}_i^t, \boldsymbol{Q}_i^t, \boldsymbol{S}_i^t\}_{i=0}^n$. Since $\boldsymbol{M}_{i,1}^t$ aims to recover the true motion between frames, $\mathcal{G}_{3D}^{t \to t-1}$ should closely align with the ground-truth Gaussians $\mathcal{G}_{3D}^{t-1}$. which can be supervised with rendering consistency. The retargeting loss is defined as:

$$\mathcal{L}_{\text{retarget}} = \sum_{i=0}^n (||\hat{\boldsymbol{I}}_i^{t \to t-1} - \hat{\boldsymbol{I}}_i^{t-1}||_2 + \lambda_{\text{SSIM}} \text{SSIM}(\hat{\boldsymbol{I}}_i^{t \to t-1}, \hat{\boldsymbol{I}}_i^{t-1}) + \lambda_{\text{LPIPS}} \text{LPIPS}(\hat{\boldsymbol{I}}_i^{t \to t-1}, \hat{\boldsymbol{I}}_i^{t-1})), \tag{6}$$

where $\hat{\boldsymbol{I}}_i^{t \to t-1} = \mathcal{R}(\mathcal{G}_{3D}^{t \to t-1}, \boldsymbol{E}_i, \boldsymbol{K}_i)$ and $\hat{\boldsymbol{I}}_i^{t-1} = \mathcal{R}(\mathcal{G}_{3D}^{t-1}, \boldsymbol{E}_i, \boldsymbol{K}_i)$ denote rendered images for view $i$, with $\mathcal{R}$ representing the rendering function for 3D Gaussian Splatting, and $\boldsymbol{E}_i$, $\boldsymbol{K}_i$ denoting camera extrinsics and intrinsics, respectively.

To ensure real-world plausibility and resolve ambiguities, we incorporate an *occlusion-aware optical flow loss* for a stronger regularization. We compute pseudo-ground-truth flow $\boldsymbol{\mu}_i^{t \to t-1}$ using SEA-RAFT (Wang et al., 2024b) and project the predicted 3D motion $\boldsymbol{M}_{i,1}^t$ to 2D scene flow as $\hat{\boldsymbol{\mu}}_i^{t \to t-1}$. A cyclic consistency mask $\mathbf{1}_{\text{cyc}}$ penalizes inconsistencies between forward and backward flows and removes occluded regions. The flow loss is defined as:

$$\mathcal{L}_{\text{flow}} = \sum_{i=0}^n \mathbf{1}_{\text{cyc}}(\boldsymbol{\mu}_i^{t \to t-1}, \boldsymbol{\mu}_i^{t-1 \to t}) \cdot ||\boldsymbol{\mu}_i^{t \to t-1} - \hat{\boldsymbol{\mu}}_i^{t \to t-1}||_2, \tag{7}$$

where $\mathbf{1}_{\text{cyc}}(\boldsymbol{\mu}_i^{t \to t-1}, \boldsymbol{\mu}_i^{t-1 \to t}) = \exp(-\boldsymbol{r}_i^t \cdot ||\boldsymbol{\mu}_i^{t \to t-1} + \boldsymbol{\mu}_i^{t-1 \to t}[\boldsymbol{p}_i^t + \boldsymbol{\mu}_i^{t-1 \to t}]||_2)$ acts as an occlusion-aware weighting term, $\boldsymbol{r}_i^t$ is a hyperparameter related to the length of each flow, and $\boldsymbol{\mu}_i^{t-1 \to t}[\boldsymbol{p}_i^t + \boldsymbol{\mu}_i^{t-1 \to t}]$ represents a pixel-wise indexing process. The *retargeting loss* and *occlusion-aware optical flow loss* are combined together as a matching supervision: $\mathcal{L}_{\text{matching}} = \mathcal{L}_{\text{flow}} + \mathcal{L}_{\text{retarget}}$.

Additionally, to naturally fuse the two sets of deformed 3D Gaussians $\mathcal{G}_{3D}^{t \to t'}$, $\mathcal{G}_{3D}^{t-1 \to t'}$ while preserving 3D Gaussians from occluded regions, we further deform the 3D Gaussian with a *dual consistency factor* $\boldsymbol{D}_i^t$ (or $\boldsymbol{D}_i^{t-1}$ for frame $t-1$). This factor is calculated by measuring the distance of the deformed concurrent 3D Gaussian point to the retrieved 3D Gaussian point at the retargeted frame via the projected 2D flow, as shown in Fig. 3. This factor serves as the guidance of areas masked in the next frame. 3D Gaussians with a factor larger than a threshold $\tau$ will be kept as occluded 3D Gaussians $\{\bar{\mathcal{G}}_{3D}^t, \bar{\mathcal{G}}_{3D}^{t-1}\}$, while the

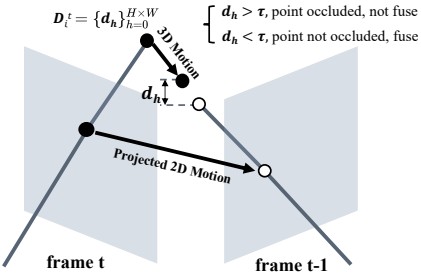

Figure 3: Dual consistency factor.

other 3D Gaussians $\{\hat{\mathcal{G}}_{3D}^t, \hat{\mathcal{G}}_{3D}^{t-1}\}$ from two nearby timestamps will be merged into one by a two-layer MLP $\boldsymbol{F}_\theta$ to eliminate temporal redundancy. The final 3D Gaussian assets $\mathcal{G}_{3D}^{t'}$ are merged with

the remaining occluded 3D Gaussians and the fused 3D Gaussians as:

$$\mathcal{G}_{3D}^{t'} = \{\bar{\mathcal{G}}_{3D}^{t \to t'}, \bar{\mathcal{G}}_{3D}^{t-1 \to t'}, \boldsymbol{F}_\theta(\hat{\mathcal{G}}_{3D}^{t \to t'}, \hat{\mathcal{G}}_{3D}^{t-1 \to t'})\}. \tag{8}$$

This fusion process is supervised by the photometric loss at novel time $t'$, which is defined as $\mathcal{L}_{\text{fusion}} = \sum_{i=0}^{n}(\|\boldsymbol{I}_i^{t'} - \hat{\boldsymbol{I}}_i^{t'}\|_2 + \lambda_{\text{SSIM}}\text{SSIM}(\boldsymbol{I}_i^{t'}, \hat{\boldsymbol{I}}_i^{t'}) + \lambda_{\text{LPIPS}}\text{LPIPS}(\boldsymbol{I}_i^{t'}, \hat{\boldsymbol{I}}_i^{t'}))$. We supervise stage 3 with the loss function $\mathcal{L}_{4D} = \mathcal{L}_{\text{matching}} + \mathcal{L}_{\text{fusion}}$.

In this way, our model achieves the task of generalized 4D human reconstruction by decomposing it into a task of static 3D Gaussian streaming and a task of dense human motion prediction. High-quality novel view images at any novel time can be acquired by interpolating the predicted dynamic 3D Gaussians and then rendering onto the corresponding image planes.

## 4 EXPERIMENT

### 4.1 EXPERIMENTAL SETTINGS

**Datasets.** *Forge4D* is trained on the DNA-Rendering (Cheng et al., 2023) training set, which comprises 2,078 human video sequences that exhibit diverse subject ages, appearances, and motion patterns. 4D synthesis is evaluated on two benchmarks: 1) an in-domain held-out test set that contains all sequences from 10 distinct identities in DNA-Rendering, and 2) the out-of-domain complete Genebody (Cheng et al., 2022) dataset. For motion prediction and metric measurement, due to a lack of ground truth annotations in real-world datasets, we construct a synthetic dataset, MetaHuman4D, with ground truth annotations for evaluation. The details of MetaHuman4D are in the Appendix B.

**Evaluation Metrics.** The synthesized image quality is measured using standard metrics: PSNR, SSIM, and LPIPS at a resolution of $518 \times 518$, unless specific ones are required by architectural constraints, such as $512 \times 512$ for GPS-Gaussian and L4GM. All models are trained and evaluated using 4 input views with a camera angle of around $45°$, which ensures sufficient coverage of the frontal human appearance. The dense motion prediction task is benchmarked using the L2 distance and the retargeted point distance. The metric scale prediction is evaluated by computing the L2 distance between predicted 3D points and their nearest corresponding points on the GT scale mesh.

**Baselines.** We establish comparisons under two experimental settings. For novel view synthesis at input timestamps, we compare against optimization-based methods (DualGS (Jiang et al., 2024), Queen (Girish et al., 2024), D-3DGS (Yang et al., 2023)) and feed-forward models (GPS-Gaussian (Zheng et al., 2024)), along with pose-free feed-forward methods (NoPosplat (Ye et al., 2024), AnySplat (Jiang et al., 2025)). For novel time interpolation quality, we compare exclusively with methods capable of generating 3D representations at non-input timestamps: SpaceTimeGS (Li et al., 2024b), D-3DGS (Yang et al., 2023), and L4DM (Ren et al., 2024).

**Implementation Details.** We initialize the backbone, camera heads, and 3D point heads using pretrained weights from VGGT. The scale head, 3D Gaussian position offset head, color offset head, scene attention blocks, and motion blocks are zero-initialized. See more details in the Appendix C.

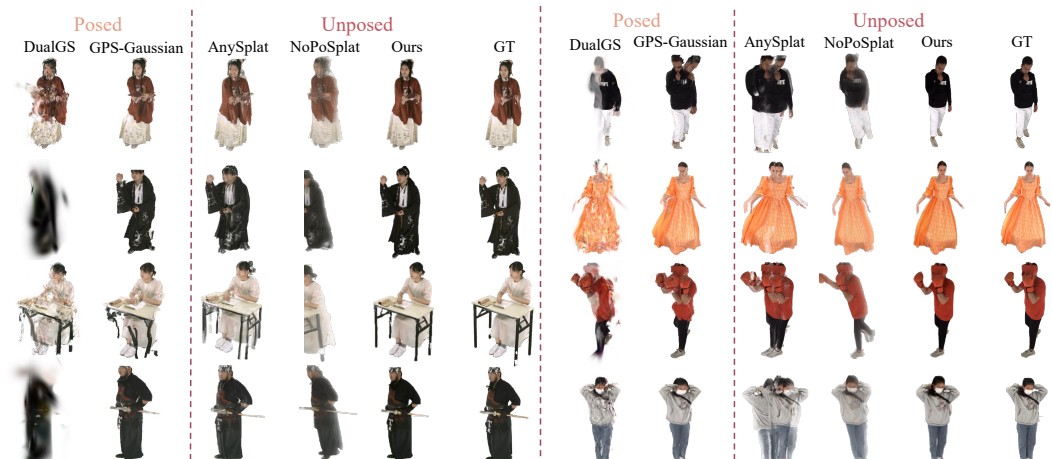

Figure 4: Qualitative results of 3D reconstruction on test sets. Our *Forge4D* exhibits more stable synthesized novel view images against artifacts, including blur, ghosting, and shape distortion.

Table 1: Quantitative results of novel-view synthesis of static reconstruction with 4 input views.

| Dataset | | | | DNA-Rendering | | | Genebody | | |
|---|---|---|---|---|---|---|---|---|---|
| Method | Type | | w. Cam. pose | PSNR↑ | SSIM↑ | LPIPS↓ | PSNR↑ | SSIM↑ | LPIPS↓ |
| DualGS | Optimization | 4D* | Yes | 18.7408 | 0.8932 | 0.1515 | 16.1721 | 0.7663 | 0.1891 |
| D-3DGS | Optimization | 4D | Yes | 20.7767 | 0.8944 | 0.1162 | 15.7067 | 0.7981 | 0.1790 |
| Queen | Streaming | 3D | Yes | 15.4966 | 0.8941 | 0.1213 | 16.4514 | 0.9484 | 0.0710 |
| GPS-Gaussian | Feed-Forward | 3D | Yes | 24.2963 | 0.9247 | 0.0867 | 25.1734 | 0.9346 | 0.0756 |
| NoPoSplat | Feed-Forward | 3D | **No** | 11.7632 | 0.8092 | 0.2846 | 13.9554 | 0.8721 | 0.1664 |
| AnySplat | Feed-Forward | 3D | **No** | 26.1157 | 0.9430 | 0.1513 | 25.8010 | 0.9287 | 0.1355 |
| Ours | Feed-Forward | 4D | **No** | **29.8167** | **0.9606** | **0.0542** | **28.0819** | **0.9523** | **0.0548** |

## 4.2 EVALUATION

**Evaluation on 3D Reconstruction and Novel View Synthesis.** As demonstrated in Tab. 1, our model outperforms all baseline methods across all metrics. Specifically, *Forge4D* surpasses previous pose-free feed-forward 3D reconstruction models, i.e., NoPoSplat and AnySplat, by up to +2.28 dB in PSNR. This significant performance gap stems not only from artifacts caused by multi-view mismatches and geometric inaccuracies but also from the inability of these methods to generate 3D models and camera configurations that align with the ground-truth scale and camera parameters. Moreover, the extensive white background in human images further complicates the synthesis of plausible renderings, even with the camera optimization procedure described in Sec. C. Compared to posed feed-forward 3D reconstruction models, *Forge4D* also exceeds the previous state-of-the-art method, GPS-Gaussian, by up to +2.90 dB in PSNR, which originates from *Forge4D*'s ability to produce more geometrically faithful reconstructions of challenging regions such as hands, heads, and accessories. Qualitative results in Fig. 4 further confirm that *Forge4D* delivers more photorealistic novel views without artifacts such as blur, ghosting, or shape distortion.

**Evaluation on 4D Reconstruction and Novel Time Synthesis.** Quantitative and qualitative results in Tab. 2 and Fig. 5 demonstrate *Forge4D*'s effectiveness in novel time synthesis. The model exhibits strong capabilities in both generating 3D Gaussian assets and interpolating 3D Gaussians for arbitrary intermediate timestamps between key frames. Our approach outperforms the previous state-of-the-art feed-forward 4D reconstruction model L4GM on both datasets with performance gains of up to +12.57 dB in PSNR. In comparison with optimization-based methods, our method surpasses all previous approaches, as these methods fail to generate reasonable novel views under such sparse-view conditions. The optimization-based techniques struggle with the limited input views, while our feed-forward approach maintains robust performance even with sparse camera arrangements.

Table 2: Quantitative results of novel-time and novel-view synthesis with 4-view videos.

| Dataset | | | DNA-Rendering | | | Genebody | | |
|---|---|---|---|---|---|---|---|---|
| Method | Type | w. Cam. pose | PSNR↑ | SSIM↑ | LPIPS↓ | PSNR↑ | SSIM↑ | LPIPS↓ |
| D-3DGS | Optimization | Yes | 20.9158 | 0.8994 | 0.1163 | 15.5823 | 0.8002 | 0.1502 |
| SpaceTimeGS | Optimization | Yes | 17.2189 | 0.8879 | 0.1247 | 14.6119 | 0.8488 | 0.1805 |
| L4GM | Feed-Forward | Yes | 18.0325 | 0.9152 | 0.1367 | 14.8572 | 0.9144 | 0.1727 |
| Ours | Feed-Forward | **No** | **29.0378** | **0.9566** | **0.0535** | **27.4247** | **0.9459** | **0.0601** |

**Evaluation on Dense Human Motion Prediction**. We evaluate our motion prediction module using our MetaHuman4D dataset containing multi-view images and dense ground-truth motions derived from mesh correspondences

Table 3: Motion prediction evaluation.

| Method | Motion Error | Point Distance |
|---|---|---|
| POMATO | 0.01274 | 0.7555 |
| Ours | **0.00953** | **0.0215** |

for each frame. The predicted 3D motions are compared against the GT 3D motions using L2 distance, and benchmarked against the state-of-the-art dense motion prediction model PO-MATO (Zhang et al., 2025c). Additionally, we report the retargeted point distance to the GT target-time mesh, where *Forge4D* consistently generates plausible outputs while POMATO fails to produce reasonable human geometry. Quantitative results in Tab. 3 demonstrate the effectiveness of our motion prediction framework. Further details are provided in Appendix B.

**Evaluation on Metric Scale Prediction**. *Forge4D* is able to recover real-world metric scale points as a byproduct of the scale prediction header supervised with *metric gauge*. We evaluate the performance of *Forge4D* in metric scale recovering by measuring the mean distance of the predicted points to the GT human mesh on MetaHuman4D, which

Table 4: Metric Scale prediction evaluation.

| Method | Point Distance |
|---|---|
| MoGe-2 | 0.3309m |
| Ours | **0.0264m** |

results in a 0.02 m error on average. A comparison with MoGe-2 is made in the same metric, with the results presented in Tab. 4. *Forge4D* outperforms MoGe-2 in mean distance to ground-truth mesh, primarily due to MoGe-2's inability to effectively align multi-view point correspondences.

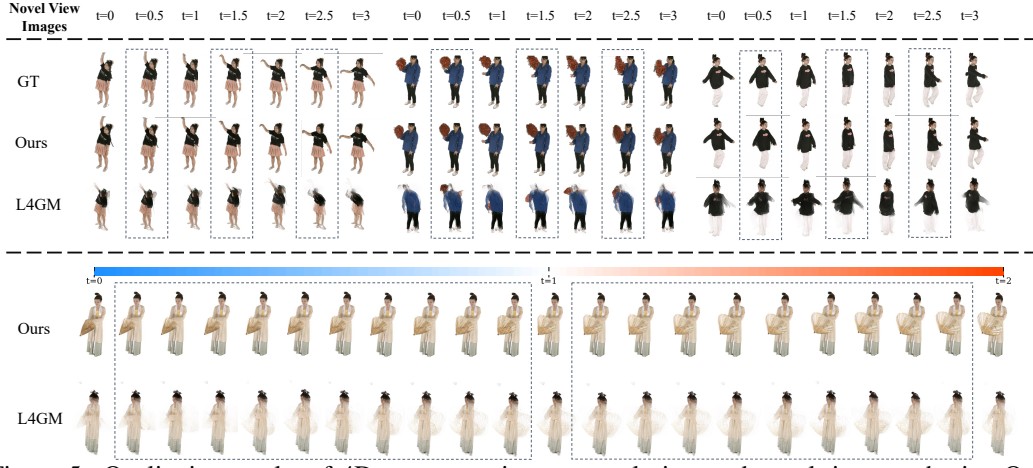

Figure 5: Qualitative results of 4D reconstruction on novel-view and novel-time synthesis. Our model accurately reconstructs 3D Gaussians for input timestamps while generating plausible intermediate 3D Gaussians *at any time* with high-fidelity rendering quality (images in dashed boxes).

## 4.3 ABLATION STUDY

**Ablations on Methodology**. We show the effectiveness of the proposed *metric gauge*, the *retargeting loss*, the *occlusion-aware optical flow loss*, and the Gaussian fusion process in Tab. 5. A significant performance gap is observed when the *retargeting loss* and the occlusion-aware *optical flow loss* are replaced with direct supervision on the novel timestamps, and the training process will lead to a collapse when the *metric gauge* alignment is missing. Additionally, directly concatenating the 3D Gaussians from the nearby 2 frames will also lead to a suboptimal novel view image quality. While the quantitative improvement in metrics may appear modest, the Gaussian fusion process plays a critical role in removing redundant 3D Gaussians and eliminating perceptually disruptive artifacts such as jittering and flickering, which are clearly visible in the video results shown in the supplementary materials.

Table 5: Ablation study on different components.

| Evaluation Mode | Variants | PSNR↑ | SSIM↑ | LPIPS↓ |
|---|---|---|---|---|
| Static Novel View | **Full Model** | 29.8167 | 0.9606 | 0.0542 |
| | w/o *gauge aligning* | 13.2884 | 0.1194 | 0.2184 |
| Dynamic Novel Time + Novel View | **Full Model** | 29.0378 | 0.9566 | 0.0535 |
| | w/o *state token* | 28.5555 | 0.9513 | 0.0592 |
| | w/o *retargeting loss* | 28.4124 | 0.9530 | 0.0573 |
| | w/o *optical flow loss* | 28.8676 | 0.9551 | 0.0563 |
| | w/o *Gaussian Fusion* | 29.0307 | 0.9556 | 0.0556 |

**Ablations on Model Speed**. We evaluate the inference speed of *Forge4D* on a NVIDIA H200 Tensor Core GPU, with the results reported in Tab. 6. The key-frame reconstruction requires 176.50 ms per inference, the motion prediction module takes 47.77 ms, and the intermediate-time interpolation for 10 frames adds 1.46 ms, resulting in a total latency of 224.27 ms per input frame pair. Given our model's capability for arbitrary-length interpolation, the effective output frame rate reaches 44 FPS when generating 10 interpolated frames per input interval.

Table 6: Ablation on model speed.

| Module | Delay | Frame Rate |
|---|---|---|
| Key-frame Reconstruction | 176.50 ms | - |
| Motion Prediction | 47.77 ms | - |
| Interpolation (10 Steps) | 1.46 ms | - |
| Full Model | 224.27 ms | 4.45 FPS |
| *+Interpolate 10 Steps* | 225.10 ms | 44.42 FPS |
| *+Interpolate 20 Steps* | 226.01 ms | 88.48 FPS |

## 5 CONCLUSION

We propose *Forge4D*, the first feed-forward model for 4D human reconstruction from uncalibrated sparse-view videos. Our approach simplifies the problem by decomposing it into two tasks: streaming real-world metric-scale 3D Gaussian reconstruction and a dense human motion prediction for novel time synthesis. *Forge4D* achieves state-of-the-art novel view synthesis quality and enables interpolation to arbitrary timestamps while maintaining plausible intermediate representations. Nevertheless, *Forge4D* exhibits certain limitations. In particular, the performance degrades in the presence of large motions or longer inter-frame intervals, primarily due to reduced inter-frame correspondences and violations of the consistent motion assumption. We identify these limitations as directions for future work, focusing on improving motion modeling under extreme displacements and optimizing computational efficiency for better interactivity.

**Ethics Statement.** The research utilizes DNA-Rendering, Genebody, and the synthesized MetaHuman4D datasets, all employing properly consented data or synthetic human models to avoid privacy concerns. Although developed for beneficial applications, we acknowledge the potential misuse of this technology for creating synthetic media without consent and encourage the development of corresponding ethical guidelines and detection mechanisms. We believe the primary impact of this research will be positive, enabling new forms of human communication and content creation, while emphasizing the need for responsible development and deployment.

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

Figure 6: Sampled examples of our MetaHuman4D test set.

## A OVERVIEW

In Section B, we detail the synthetic MetaHuman4D dataset, including its composition and our ground-truth motion annotation methodology. Section C provides additional training and evaluation specifics, along with visualizations of *Forge4D*'s motion prediction results. Further ablations concerning video duration, temporal intervals, and camera count are presented in Section D. Section E illustrates *Forge4D*'s metric scale prediction capabilities. Section F states how LLM participates in this work.

## B OUR SYNTHESIS METAHUMAN4D DATASET

Since there are no ground truth annotations of dense human motion and metric scale in the captured real-world dataset, we construct a synthesis dataset with ground truth annotations for the evaluation of different methods on these two tasks.

**Dataset details.** Our synthesized test set contains 11 different identities and 7 motion types. For each person, we select a motion sequence to animate it and render dynamic videos from 48 views with the Unreal rendering engine and save the GT mesh model of the human at each given timestamp. The diversity of this dataset is at the same scale as the commonly used test set from DNA-rendering, which covers 10 identities. We visualize some samples in Fig. 6.

**Ground truth motion annotation.** We derive ground-truth motion from sequential human meshes by computing the displacement of corresponding points between consecutive timestamps. Specifically, for a point $\bar{x}_i^t$ on the mesh at time $t$, we obtain its backward motion as $\bar{m}_1^t = \bar{x}_i^{t-1} - \bar{x}_i^t$ and its forward motion as $\bar{m}_2^t = \bar{x}_i^{t+1} - \bar{x}_i^t$. During evaluation, we establish correspondences between predicted points and ground-truth mesh points, then compute the motion error between the predicted motions $m_{1,2}^t$ and the ground-truth motions $\bar{m}_{1,2}^t$ for each matched point pair.

## C IMPLEMENTATION DETAILS

**Training detail of the three training stages.** Stage 1 employs a learning rate initialized at $5 \times 10^{-5}$ for the Gaussian and scale heads, and $1 \times 10^{-5}$ for other components. Stage 2 initializes with Stage 1 weights, except for the randomly initialized state attention module and state tokens. During this stage, the state attention module, state tokens, scale head, Gaussian head, and pose head are optimized with a $5 \times 10^{-5}$ initial learning rate, while other components remain frozen. Stage 3 initializes with Stage 2 weights, with randomly initialized motion blocks, Gaussian fusion, and

zero initialized motion head module, all trained at $5 \times 10^{-5}$ initial learning rate, while all other parameters remain frozen. All stages use a consistent batch size of 8 and are trained on 8 H20 GPUs for 100,000 iterations each, and the learning rate is linearly decreasing to $1 \times 10^{-5}$ at the end of the training stage. Hyperparameters are empirically set as: $\lambda_{\text{SSIM}} = 0.25$, $\lambda_{\text{LPIPS}} = 0.25$, $\tau = 0.05$, $\boldsymbol{r}_i^t = 0.1 \cdot ||\boldsymbol{\mu}_i^{t \to t-1}||_{2,2} + 0.5$. $|| \cdot ||_{2,2}$ represents the L2 norm on the second dimension of the tensor.

**Metric scale recovery during inference.** During inference, metric-scale 3D points are recovered by dividing the output 3D points by the predicted metric gauge $\hat{p}_{\text{gauge}}$, with the corresponding adjustments applied to the scale and motion attributes of 3D Gaussians for novel-view and novel-time rendering.

**Evaluation of pose-free methods.** Same with previous works (Ye et al., 2024; Fan et al., 2024; Wang et al., 2021), we optimize novel view camera positions while keeping the 3D Gaussians fixed for pose-free methods (NoPosplat (Ye et al., 2024), AnySplat (Jiang et al., 2025), *Forge4D*) to address the inherent ambiguity that multiple 3D configurations can explain the same 2D observations. Importantly, this optimization is solely performed for evaluation and is not required during actual deployment for novel-view and novel-time rendering.

**Evaluation details of different baselines on novel-view synthesis of static reconstruction.** In Tab. 1, all 3D methods are evaluated using frame-wise reconstruction and rendering, while DualGS and Queen are optimized using full video sequences. We disable state tokens and state attention blocks in Forge4D for fair comparisons. Note that DualGS cannot synthesize novel-time 3D scenes.

**Evaluation details of different baselines on novel-time and novel-view synthesis.** All 4D methods are evaluated at a sampling rate of 2, which is holding out 1 timestamp between every 2 input timestamps for novel-time and novel-view evaluation. Since L4GM can only take a maximum input length of 8 timestamps due to the GPU memory limitation, we compare it with our model at a same input video length of 8 timestamps for fair comparison. However, we claim that our model can be extended to arbitrary length of videos without suffering from memory accumulation issues, thanks to the effectiveness of the state token embeddings.

**Evaluation details of dense motion prediction.** For our experimental setup, we select 4 sparse views from the rendered videos as input to each compared method for human reconstruction. To ensure a fair quantitative evaluation, we first align the predicted 3D points at the initial timestamp with the ground-truth mesh from the MetaHuman4D dataset using scaling, translation, and rotation transformations. This alignment step eliminates errors arising from scale and coordinate system discrepancies, especially for POMATO. Subsequently, for each predicted 3D point, we identify the closest point on the ground-truth mesh, establishing a correspondence for motion evaluation. The motion accuracy is quantified by computing the L2 distance between the motion vectors of each predicted point and its matched ground-truth point. Additionally, we report the mean distance between the deformed predicted 3D points and their corresponding ground-truth points at the target timestamp. We visualize the predicted 3D points and sampled motions in Fig. 7.

**Evaluation details of metric measurement.** We evaluate *Forge4D* and MoGe-2 (Wang et al., 2025b) using identical videos with dense motion annotations. For each predicted 3D point, we compute its minimum distance to the ground-truth mesh surface as the per-point prediction error. The mean distance is calculated directly without additional alignment procedures, beyond the necessary transformation of predicted points from camera coordinates to world coordinates. For MoGe-2, we independently calculate the prediction error for each view, and make an average on 4 input views.

## D MORE ABLATIONS

**Ablations on Video Duration**. We evaluate our model under different video durations and report the metrics of novel view and novel time image quality, in terms of PSNR, SSIM, and LPIPS. The results in Tab. 7 show that the duration of the video has a limited impact on the model performance.

Table 7: Ablation on the video length.

| Timestamps | 8 | 16 | 32 |
|---|---|---|---|
| PSNR↑ | 29.0378 | 29.3932 | 29.3615 |
| SSIM↑ | 0.9566 | 0.9591 | 0.9591 |
| LPIPS↓ | 0.0535 | 0.0519 | 0.0519 |

**Ablations on Time interval and Interpolation rate**. *Forge4D* is trained by sampling a timestamp every two timestamps as novel time supervision frames, which we refer to as a sampling rate of two.

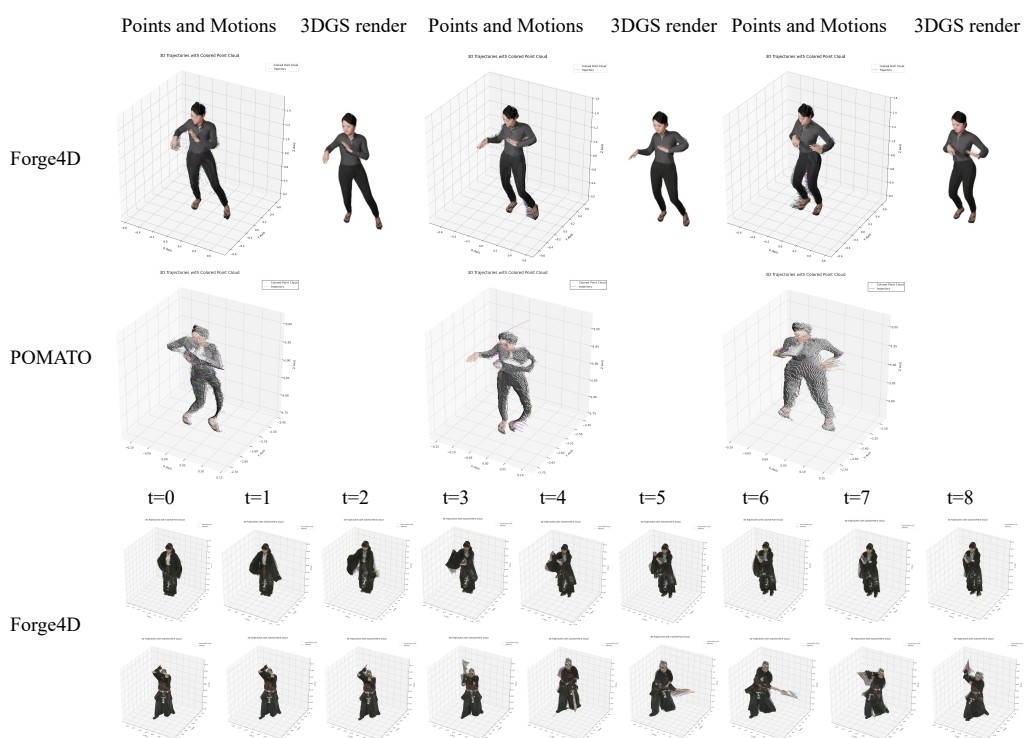

Figure 7: Dense motion prediction visualization. Zoom in to see more details.

We show that this supervision strategy is sufficient and that *Forge4D* can generalize to longer time intervals. In addition, more plausible intermediate frames can be acquired by adjusting the interpolation rate. We carried out metric calculations for larger sampling rates of 4 and 8. The quantitative result in Tab.8 indicates that the rendered novel view quality is preserved even when a longer duration is held between the two input timestamps. There is only a reasonable drop in PSNR of at most 3.04 dB, which is mainly because the linear velocity assumption is hard to maintain under a longer duration.

Table 8: Ablation on the sampling rate.

| Sampling Rate | 2 | 4 | 8 |
|---|---|---|---|
| PSNR↑ | 29.0378 | 27.9129 | 25.9673 |
| SSIM↑ | 0.9566 | 0.9538 | 0.9344 |
| LPIPS↓ | 0.0535 | 0.0604 | 0.0767 |

**Ablations on Camera number**. Although *Forge4D* is trained with a consistent configuration of 4 input views, the model demonstrates strong generalization capability to arbitrary numbers of input cameras. As shown in Table 9, which evaluates novel view and novel time synthesis quality with 2, 4, and 5 input views, the performance degradation remains minimal, with a maximum decrease of only 1.90 dB in PSNR. This robustness stems from the inherent stability of our backbone architecture and the fact that cross-frame information aggregation is largely decoupled from the final 3D Gaussian prediction heads, allowing the model to adapt effectively to varying numbers of input views.

Table 9: Ablation on camera settings.

| Cam. Num. | 2 | 4 | 5 |
|---|---|---|---|
| PSNR↑ | 27.1316 | 29.0378 | 28.1337 |
| SSIM↑ | 0.9384 | 0.9566 | 0.9471 |
| LPIPS↓ | 0.0648 | 0.0535 | 0.0613 |

## E METRIC MEASUREMENT

We provide additional human body metric measurement results in Figure 8, demonstrating that *Forge4D* achieves accurate metric-scale reconstruction across both synthetic datasets and real-world captures, validating its robustness in practical measurement applications.

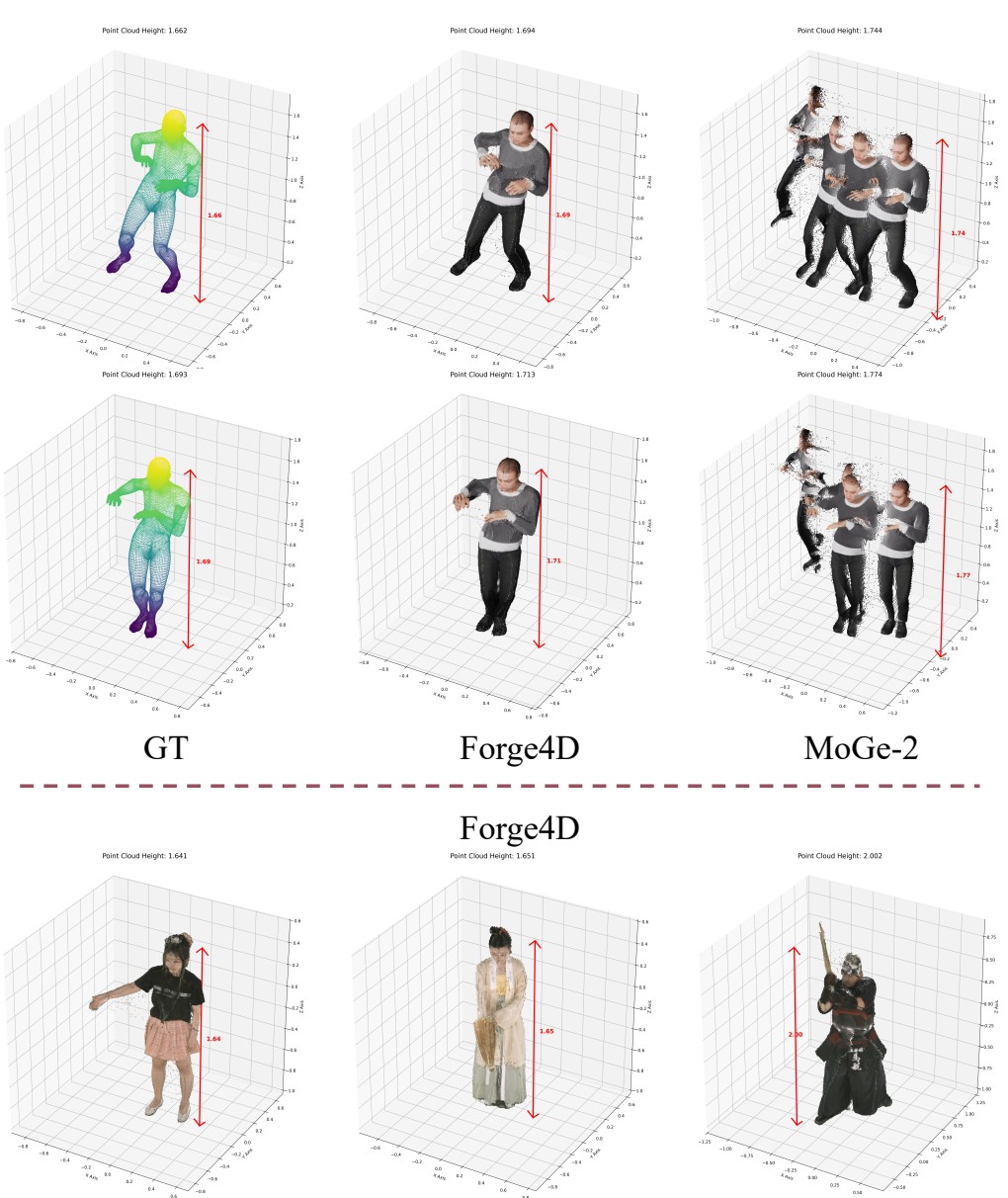

Figure 8: Samples of points prediction in metric scale.

## F    STATE OF LLM USAGE

We use LLM to assist in coding and paper polishing. There is no further use of LLM for the idea formulation, experiments, and main paper writing.

