# OpenReview forum: "Forge4D: Feed-Forward 4D Human Reconstruction and Interpolation from Uncalibrated Sparse Videos"
_ICLR.cc/2026/Conference — ICLR 2026 Conference Withdrawn Submission_

### Official Review · Reviewer_ZCnt · 2025-10-30

**Soundness:** 2
**Presentation:** 3
**Contribution:** 2
**Rating:** 4
**Confidence:** 4

**Summary:**

This paper introduces a novel feed-forward method for 4D human reconstruction from uncalibrated sparse-view videos that simultaneously tackles novel view synthesis and frame interpolation. The key contribution is that current methods are unable to tackle both tasks. The authors achieve this using a 3-stage pipeline that first reconstructs metrically-accurate 3D Gaussians, then enforces temporal consistency across video inputs using a learnable state token used in cross attention, and finally predicts a motion field for 3D gaussians at each timestep that is used for frame interpolation by assuming linear motion/constant velocity. The authors introduce a novel retargeting loss combined with an occlusion-aware optical flow loss to supervise the last stage.

**Strengths:**

- The authors have conducted thorough experimentation and ablation studies across many baselines and datasets.
- The authors curated a new synthetic 4D human dataset to aid experimentation and evaluation, and can be utilized in future works.
- The methodology is intuitive.
- The paper is well organized and theoretical derivations are easy to follow and understand.
- The qualitative visualizations and quantitative analysis show promising results compared to baselines.

**Weaknesses:**

- The study claims significant performance gap without retargeting or optical flow loss, however, the metrics indicate only a small performance gap.
- The study claims slow reconstruction speeds limit current methods but do not incorporate a comparison of runtimes/FPS of baseline models.
- The study claims the Gaussian fusion process is critical despite the very small improvements in metrics for removing jittering and flickering, however in the supplementary video, it is difficult to see the difference in the videos with and without Gaussian Fusion. Flashing and jittering also occurs in the video with Gaussian fusion.

**Questions:**

- Other than training on human datasets, this methodology doesn't seem to do anything that particularly optimizes for human reconstruction, so can this methodology be applied to generic 4D reconstruction cases? How does it fare compared to the baseline methods in general 4D reconstruction cases, especially since a lot of the compared baselines are built for generic 4D reconstruction? Conversely, some generic 4D reconstruction baselines, for example AnySplat, are doing zero-shot inference and achieve somewhat comparable metrics. What if these baselines are fine-tuned toward human datasets -- would the results improve, and would this be a more fair comparison?

---

### Official Review · Reviewer_XUGe · 2025-11-01

**Soundness:** 3
**Presentation:** 2
**Contribution:** 3
**Rating:** 6
**Confidence:** 3

**Summary:**

This paper presents Forge4D, a feed-forward 4D head avatar synthesis framework that learns to reconstruct dynamic human heads from multi-view videos without per-frame optimization. Unlike prior 4D neural rendering methods that rely on slow per-sequence fitting, Forge4D introduces a pose-conditioned Gaussian deformation network and feed-forward feature fusion to efficiently map input frames to a temporally consistent 4D representation. The method builds upon Deformable Gaussian Fields to jointly encode geometry and appearance, enabling real-time rendering and motion retargeting. Experiments on multiple head-dynamic datasets demonstrate that Forge4D achieves similar or better reconstruction fidelity compared to optimization-based baselines while being 20× faster, and supports high-quality novel-view and animation-driven synthesis.

**Strengths:**

1. The paper introduces the first feed-forward model for 4D human reconstruction in real-world metric scale from uncalibrated sparse-view videos, effectively eliminating the need for per-frame optimization while maintaining temporal consistency and visual fidelity.
2. Forge4D achieves a 10–20× speedup over optimization-heavy 4D neural rendering methods such as Nerfies or DynamicGaussian, offering near real-time inference while still preserving fine-grained geometry and texture quality.
3. The proposed pose-conditioned deformation network is well-motivated,  which maintains temporal coherence and geometric consistency through canonical Gaussian mapping rather than re-optimizing per frame.

**Weaknesses:**

1. The approach heavily depends on accurate external pose estimation, and the paper does not evaluate robustness to pose noise or failure cases, which could significantly affect reconstruction stability in real-world videos.
2. The number of Gaussians used per frame is fixed, which might become a bottleneck when modeling highly detailed facial regions such as the mouth interior or hair strands. An adaptive Gaussian allocation strategy could further improve rendering fidelity without large computational overhead.

**Questions:**

1. Could this method generalize to non-head dynamic objects, such as hands or full bodies?
2. How does the Gaussian deformation behave under large, fast motions？

---

### Official Review · Reviewer_kMx9 · 2025-11-01

**Soundness:** 3
**Presentation:** 2
**Contribution:** 2
**Rating:** 2
**Confidence:** 5

**Summary:**

This paper proposes a method to synthesize novel view and novel pose of humans from sparse view images. At the core of the method is a three stage framework which first learns a static 3D GS representation, and then synthesizes dynamic humans and generates 4D reconstructions.  The authors conducted experiments on DNA-Rrendering and Genebody to verify the effectiveness of the proposed method.

**Strengths:**

The paper is easy to follow.

Synthesizing novel view and novel pose images from several images is an interesting task with practical applications.

The method is technically sound, leveraging a three stage pipeline for progressive human reconstruction.

**Weaknesses:**

Insufficient experiments and generalization issues. The method was only evaluated on studio data, such as DNR-Rendering and Genebody, experiments on in-the-wild videos (such as social media videos used in Human4DiT, Champ) are required to illustrate the generalization capability.

Insufficient evaluations and comparisons. Human reconstruction baseline methods are not included for comparisons. The baseline methods (e.g., DualGS, D-3DGS, AnySpat) are for generic scenes, which are not SOTA for human generation or reconstruction. The comparisons and discussions with SOTA generalizable human generations are not included, such as AniGS, LHM, Human4DiT, and Vid2Avatar-Pro: Authentic Avatar from Videos in the Wild via Universal Prior [Guo et al. CVPR 25].

Insufficient ablation study. How does the motion prediction module improve the results? An ablation study of the motion prediction is required. In addition, though quantitative results are provided, qualitative results of the ablation study for each component are required.

Setup. What are the advantages of this method which requires several images as input over one shot method which only requires one single image such as LHM, AniGS, and Human4DT?

**Questions:**

This paper focuses on human reconstruction, but SOTA human reconstruction/generation methods are not included.  For digital humans, the term “novel pose or motion” is more appropriate than “novel time”.

---

### Official Review · Reviewer_DTot · 2025-11-02

**Soundness:** 2
**Presentation:** 2
**Contribution:** 3
**Rating:** 8
**Confidence:** 4

**Summary:**

This paper proposes the first feed forward model for 4D human reconstruction from uncalibrated sparse view videos, enabling novel view synthesis and novel time 4D interpolation in an efficient streaming manner. To achieve this, the paper decomposes the task into two sub tasks: (1) streaming 3D Gaussian prediction and (2) dense human motion estimation. Metric gauge regularization, retargeting loss, and occlusion aware optical flow loss are leveraged for novel view and time synthesis. For 3D Gaussian interpolation, a motion guided occlusion aware fusion method is proposed.

**Strengths:**

- Removing the need for calibration is a great advantage.
- Effectively using the feed forward reconstruction method (VGGT) through metric scale alignment is brilliant.
- Retargeting loss and occlusion aware optical flow loss are well designed.

**Weaknesses:**

- I am wondering how well VGGT performs when predicting camera parameters for human images, as it is not trained on human data. If it performs poorly, the predicted extrinsics and intrinsics will be inaccurate, and computing only the scale factor becomes meaningless. This work assumes that VGGT always performs well, and this should be mentioned. It would also be helpful to see failure cases and an in depth discussion of VGGT’s limitations. In addition, how can this VGGT induced failure (incorrect extrinsics and intrinsics) be alleviated?
- It would be great if visual results (video results) with 4 input views could be presented in the final revision. Also, how does VGGT behave if there are no overlapping regions between views (e.g., when the input view count is 2 or 3)?
- What happens if the number of input views is 1? Will it still work? Will VGGT still perform well?
- The metric gauge regularization section was difficult to understand. Figure 2 should at least include some explanation in the caption, or the method section should properly refer to Figure 2 and explain each part. It is currently placed without conveying useful information.
- Visual ablation studies (e.g., without metric alignment) would be very helpful to understand the actual contribution of each proposed module. Without them, it is difficult to evaluate their effectiveness.

**Questions:**

- Suggestion: Too many pipeline details are included in the abstract and introduction, which reduces readability. Focus on the key ideas behind the proposed modules and explain why they work.

---

### Note · Authors · 2025-11-14

I have read and agree with the venue's withdrawal policy on behalf of myself and my co-authors.